# Mitigating Multimodal Hallucinations via Gradient-based Self-Reflection

## Abstract

Hallucination in Multimodal Large Language Models (MLLMs) occurs when inaccurate text-visual alignments are generated, posing a major challenge for reliable model output. Previous studies have identified three primary biases as major causes of hallucinations: text-visual bias (over-reliance on text over visual details), co-occurrence bias (misleading object correlations), and long-term bias (increased hallucinations in later stages of long sequences). Existing hallucination mitigation methods often rely on visual grounding, which requires additional resources such as scoring systems using another MLLM, and still fail to fully address all biases, particularly co-occurrence bias in visual inputs. We propose Gradient-based Influence-Aware Contrastive Decoding (GACD) to explicitly and jointly balance these biases, thereby mitigating hallucinations. To quantify these biases at the individual sample level, we introduce 'token influence'. Since biases are rooted in the training data and become embedded in pre-trained MLLMs, we derive token influence through self-reflection by calculating the gradients from output predictions to input tokens. Notably, GACD is the first approach capable of fully addressing co-occurrence bias without relying on extra resources or any form of tuning. Extensive experiments demonstrate GACD's effectiveness in reducing hallucinations and improving MLLM performance, achieving new state-of-the-art results while providing insights into the visual perception capabilities of these models.

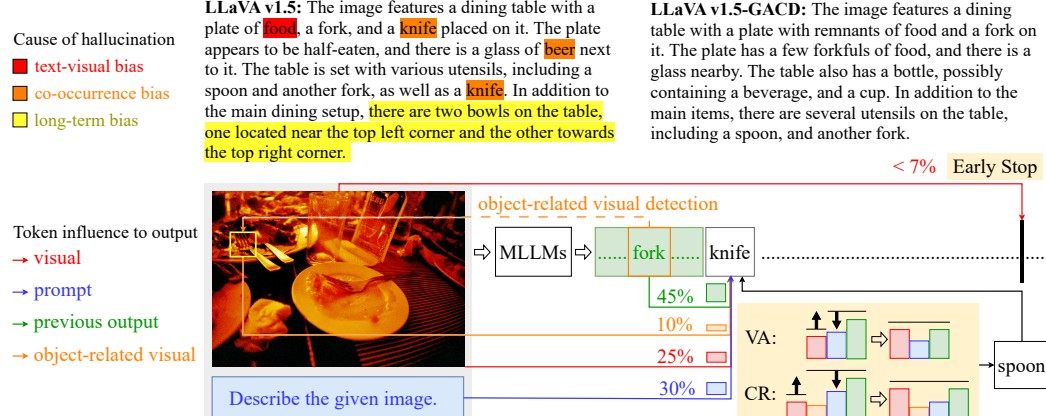

Figure 1: An illustration of our proposed GACD method and its effectiveness in addressing hallucinations caused by three key biases: text-visual bias, co-occurrence bias, and long-term bias. GACD measures token influence via gradients to uncover biases embedded in pre-trained MLLMs. Specifically, **GACD-VA** balances text-visual bias by increasing the visual influence to match the influence from the prompt or the previous output. **GACD-CR** detects distracting-object-related visual tokens and amplifies the influence of visual tokens unrelated to these distracting objects, effectively mitigating co-occurrence hallucinations. Additionally, an **early stopping** strategy based on visual influence is introduced to prevent long-term hallucinations.

# 1 INTRODUCTION

Multimodal Large Language Models (MLLMs) have recently gained significant momentum due to their ability to process and generate text from various data modalities, including images and videos. Despite their impressive capabilities, these models face a significant challenge: hallucination, where the generated text content is not grounded in the visual inputs.

Hallucinations in MLLMs arise from three key biases extensively discussed in the literature Li et al. (2023); Kang & Choi (2023); Kim et al. (2024): text-visual bias, co-occurrence bias, and long-term bias. **Text-visual bias** arises when a model relies too heavily on textual input, neglecting visual information. **Co-occurrence bias** results from statistical correlations in training data, leading models to predict objects based on frequent co-occurrence, even when absent from visual inputs. For instance, in a LLaVA-1.5 generated description (see Fig. 1), 'knife' and 'beer' are described, while 'fork' and 'glass' appear in the input image. **Long-term bias**, highlighted in yellow in Fig. 1 and analyzed as 'position factors' in Zhou et al. (2023); Favero et al. (2024), refers to an increased likelihood of hallucinations in later stages of long-form sequences, often caused by earlier predictions overly influencing subsequent content generation.

Existing methods mostly focus on mitigating visual-text bias by grounding predictions in visual inputs. These methods typically require additional resources to check predictions, such as scoring systems that involve another MLLM prone to hallucination Xing et al. (2024); Deng et al. (2024); Radford et al. (2021), or use contrastive decoding Leng et al. (2023); Favero et al. (2024) without precise mechanisms to balance bias. On the other hand, because these approaches ground predictions on visual inputs alone, they fail to eliminate distractions caused by irrelevant or misleading visual information, thus co-occurrence bias remains a problem. Long-term bias is usually addressed in the literature by incorporating sequence length as a model parameter Zhou et al. (2023); Favero et al. (2024), capturing only a coarse likelihood of hallucination based solely on sequence length. Among all these existing methods, bias mitigation often depends on fixed hyperparameters uniformly applied across all samples, requiring costly tuning and would underperform considering the fact that biases behave differently for each single sample. There are also methods that mitigate bias by training Chen et al. (2023); Jiang et al. (2024); Yue et al. (2024); Ben-Kish et al. (2023); Sun et al. (2023) and reinstructing the model through fine-tuning, which requires a huge cost on data collection and model training. Therefore, how to jointly and reliably analyze and address all these biases free from additional training or inputs remains an unresolved and challenging issue.

In this work, we propose Gradient-based Influence-Aware Contrastive Decoding (GACD) to solve the problem. We first introduce a mathematical examination termed 'token influence' to reflect the sensitivity between output and input tokens (visual, prompt, previous output in Fig. 1), using gradient norms. This sample-wise, gradient-based influence reveals the local linear behavior of MLLMs, showing that small changes in input lead to proportional changes in output, thereby exposing inherent biases embedded in the model. We then mitigate co-occurrence hallucinations, the first one addressing this type of hallucination to the authors' knowledge, by amplifying the influence of visual tokens unrelated to distractions (GACD-CR), and reduce visual-text hallucinations by amplifying all visual input (GACD-VA). Additionally, to address long-term bias, we introduce a precise stopping criterion that halts generation when the visual influence ratio falls below a set threshold.

To be specific, we find that hallucinations are largely driven by the ratio of visual token influence relative to the influence of all input tokens. This ratio serves as the underlying factor linking text-visual bias, co-occurrence bias, and long-term bias, thereby explaining how these biases contribute to hallucinations. To address co-occurrence bias, GACD-CR identifies the visual tokens associated with the distracting object and labels it as a distracting-object-related token. For example, "fork" is a distracting object, and its related visual tokens are detected in Fig. 1. We categorize visual tokens into two groups: those related to the distracting object and those unrelated. We then employ influence-aware contrastive decoding to explicitly enhance the influence of essential visual tokens and reduce the impact from distracting visuals to outputs. In GACD-VA, we amplify the influence of all visual tokens (only those unrelated to the distracting object when combined with CR). This is done to ensure that the influence of these tokens matches the maximum influence of other components, establishing it as the dominant influence to mitigate the text-visual bias, as illustrated in the lower-right part of Fig. 1.

We summarize our contributions as follows:

- We propose GACD, a novel hallucination mitigation method that enhances the influence of essential visual tokens and jointly addresses text-visual, co-occurrence, and long-term bias via gradient-based self-reflection, without requiring external resources, training, or tuning.

- GACD is the first method capable of fully addressing co-occurrence bias, even when distracting objects present within the visual inputs.

- We propose token influence and formulate bias problems of MLLMs through mathematical expressions, enabling a detailed understanding of biases at the individual sample level.

- GACD significantly improves baseline MLLMs, establishing new SOTA results. It achieves up to a 6.2 F1 boost on POPE Li et al. (2023) and up to a 2.97 accuracy increase on the LLaVA-QA90 Liu et al. (2024b) across multiple MLLMs.

## 2 RELATED WORK

**Hallucination and Bias**. Wang & Sennrich (2020) demonstrated that hallucinations are more pronounced in out-of-domain test sets. As noted by Tonmoy et al. (2024); Li et al. (2023); Fu et al. (2024), hallucinations are closely related to biases, particularly text-visual and co-occurrence bias. Long-term bias, inherited from LLMs, has been studied in Favero et al. (2024); Zhou et al. (2023). Existing methods Li et al. (2023); Fu et al. (2024); Kim et al. (2024) typically report only overall statistics, lacking a mathematical sample-wise bias analysis. This distinction is crucial, as biases can vary from case to case. We argue that biases are embedded in the parameters of pre-trained MLLMs, as they inherently reflect the biases present in the training dataset. Therefore, analyzing sample-wise bias through self-reflection offers a straightforward approach.

**Hallucination Mitigation**. Existing hallucination mitigation methods can be grouped into two categories: training-related and inference-related methods. Training-related methods Chen et al. (2023); Jiang et al. (2024); Yue et al. (2024); Ben-Kish et al. (2023); Sun et al. (2023) are expensive, requiring access to training data and specialized modifications to that data to effectively mitigate hallucinations. Inference-related methods are further divided into revision methods and guided or contrastive decoding methods. Revision methods Xing et al. (2024); Deng et al. (2024); Radford et al. (2021); Zhai et al. (2024) often use additional scoring systems, typically involving another MLLM, which itself may hallucinate. In contrast, guided or contrastive decoding methods Leng et al. (2023); Zhao et al. (2024); Favero et al. (2024), adjust logits during inference without extra training, making them easily integrable as add-on modules to any MLLM. Our approach falls into this category. However, existing decoding methods still rely on extra resources. For instance, Leng et al. (2023) utilizes noise images for contrastive decoding, while Zhao et al. (2024) employs a visual decoder for guided decoding. Furthermore, their decoding weights are fixed using hyperparameters Leng et al. (2023); Zhao et al. (2024) or adjusted based on the length of the generated text Favero et al. (2024). In contrast, our method does not rely on any external resources and tuning. Instead, it employs gradient-based influence analysis to calculate precise, sample-specific decoding weights. This approach accurately balances multiple biases, leading to improved hallucination mitigation.

## 3 BIASES IN MULTIMODAL LARGE LANGUAGE MODELS

In this section, we analyze three key biases that underlie hallucinations in MLLMs: text-visual bias, co-occurrence bias, and long-term bias. We begin by introducing MLLMs, followed by a discussion on measuring the influence of input tokens, and conclude with an in-depth analysis of each bias.

**Introduction to MLLMs**. Consider a sequence $\mathbf{t}^p = [t_1^p, \ldots, t_N^p]$ as the input prompt, where $t_n^p$ ($1 \leq n \leq N$) is a prompt token from a predefined vocabulary. Visual tokens $\mathbf{t}^v = [t_1^v, \ldots, t_S^v]$ are extracted from visual inputs $V$, using $\mathbf{t}^v = \mathcal{E}_\tau(V)$, where $\mathcal{E}_\tau(\cdot)$ is a visual encoder. These tokens are mapped to the token space via linear projection. The MLLMs generate the response sequence $\mathbf{y} = [y_1, \ldots, y_M]$ using the logit generation function $\mathcal{F}_\theta(\cdot)$ and the softmax function $\sigma(\cdot)$ [1] as:

$$\mathbf{y} = \sigma(\mathcal{F}_\theta(\mathbf{t}^v, \mathbf{t}^p)), \tag{1}$$

---

[1]The output probability of the softmax function is interpreted as confidence in this paper.

where $y_m$ denotes an individual output token for $1 \leq m \leq M$. The conditional probability distribution $p(\mathbf{y}|\mathcal{F}_\theta, \mathbf{t}^p, \mathbf{t}^v)$ can therefore be expressed as:

$$p(\mathbf{y}|\mathcal{F}_\theta, \mathbf{t}^v, \mathbf{t}^p) = \prod_{m=1}^{M} p(y_m|\mathcal{F}_\theta, \mathbf{t}^v, \mathbf{t}^p, \mathbf{y}_{<m}), \tag{2}$$

where $\mathbf{y}_{<m} = [y_1, \ldots, y_{m-1}]$ for $m > 1$ and is empty for $m = 1$. The pre-trained MLLMs ($\theta^\star$ and $\tau^\star$) are trained to minimize the overall loss of the entire training data:

$$\tau^\star, \theta^\star = \arg\min_{\tau,\theta} \mathcal{L}(\sigma(\mathcal{F}_\theta(\mathcal{E}_\tau(V), \mathbf{t}^p)), \mathbf{y}^{gt}), \tag{3}$$

where $\mathcal{L}(\cdot)$ is the loss function of MLLMs, and $\mathbf{y}^{gt}$ denotes the ground truth outputs. During training, MLLMs learn not only factual knowledge but also statistical correlations and biases present in the data, which become embedded in the model's parameters. As a result, these biases are captured within the pre-trained parameters. We aim to analyze these biases by analyzing the parameters $\tau^\star$.

**Token Influence Measurement**. To quantify token influence, we analyze the behavior of the model function $\mathcal{F}_{\theta*}(\mathbf{t}^v, \mathbf{t}^p)$ for prediction at the $m^{\text{th}}$ output, focusing on the local region around the visual embedding $\mathbf{t}^{v(0)}$ and the text prompt $\mathbf{t}^{p(0)}$. The linear behavior in the local area is likely to contain information about the preference. A straightforward approach is to consider the first order Taylor expansion [2] of $\mathcal{F}_{\theta*}$ w.r.t $\mathbf{t}^v, \mathbf{t}^p$ and, all the previous output tokens $\mathbf{y}_i$ with $i < m$:

$$\mathcal{F}_{\theta*}(\mathbf{t}^v, \mathbf{t}^p)_m \approx \sum_{s=1}^{S} \mathbf{g}_{ms}^v \cdot t_s^v + \sum_{n=1}^{N} \mathbf{g}_{mn}^p \cdot t_n^p + \sum_{i=1}^{m-1} \mathbf{g}_{mi}^y \cdot y_i + Const, \tag{4}$$

where:

$$\mathbf{g}_{ms}^v := \left.\frac{\partial(\mathcal{F}_{\theta*})_m}{\partial t_s^v}\right|_{\mathbf{t}^v = \mathbf{t}^{v(0)}}, \quad \mathbf{g}_{mn}^p := \left.\frac{\partial(\mathcal{F}_{\theta*})_m}{\partial t_n^p}\right|_{\mathbf{t}^p = \mathbf{t}^{p(0)}}, \quad \mathbf{g}_{mi}^y := \frac{\partial(\mathcal{F}_{\theta*})_m}{\partial y_i} \tag{5}$$

are the first order gradients terms on visual tokens $\mathbf{t}^v$, text prompt $\mathbf{t}^p$ and previous outputs $y_i$, $Const$ denotes all other terms that are constant w.r.t the $\mathbf{t}^v, \mathbf{t}^p$.

In this sense, $\mathcal{F}_{\theta*}(\mathbf{t}^v, \mathbf{t}^p)$ can be seen as a linear classifier w.r.t the visual tokens, text prompt and the previous outputs around a certain data point $(\mathbf{t}^{v(0)}, \mathbf{t}^{p(0)})$. The influence of tokens is indicated by the weight norm of the linear classifier, specifically, the gradient norm. We choose the Manhattan (L1) norm for our analysis and experiments because its sparsity clearly identifies the contribution of individual tokens. Consequently, the influence of each token can be represented as $\left|\mathbf{g}_m^{v/p/y}\right|$, and the overall influence of visual tokens, text prompt, and previous outputs can be represented as:

$$\mathtt{I}^v := \sum_{s=1}^{S} \left|\mathbf{g}_{ms}^v\right|, \quad \mathtt{I}^p := \sum_{n=1}^{N} \left|\mathbf{g}_{mn}^p\right|, \quad \mathtt{I}^y := \sum_{i=1}^{m-1} \left|\mathbf{g}_{mi}^y\right|. \tag{6}$$

This token influence expression allows us to conduct a mathematical analysis of sample-wise biases.

**Text-visual Bias** occurs when the influence of the text prompt tokens significantly outweighs that of the visual tokens, which can be expressed as $\mathtt{I}^p \gg \mathtt{I}^v$ or $\mathtt{I}^y \gg \mathtt{I}^v$, according to our proposed components influence. We begin our analysis of this bias with an object existence Visual Question Answering (VQA) task, involving 'yes/no' predictions, by calculating the overall influence ratio:

$$r^v = \frac{\mathtt{I}^v}{\mathtt{I}^v + \mathtt{I}^p + \mathtt{I}^y}, \quad r^p = \frac{\mathtt{I}^p}{\mathtt{I}^v + \mathtt{I}^p + \mathtt{I}^y}, \quad r^y = \frac{\mathtt{I}^y}{\mathtt{I}^v + \mathtt{I}^p + \mathtt{I}^y}, \tag{7}$$

on the POPE dataset Li et al. (2023). In this case, $\mathtt{I}^y$ is always zero, as the answer is a single word based on the question. The blue bars in Fig. 5a, which display visual influence ratio across LLaVA-v1.5, InstructBLIP, and mPLUG-Owl2, show that the prompt input exerts more influence than the visual tokens. This phenomenon is common in MLLMs and can be attributed to their training process, where multimodal features are aligned with language tokens after extensive text-based pre-training, causing language components to dominate the decision-making process.

We further observe text-visual bias in the open-ended image captioning task (Fig. 2) and the VQA task (Fig.1 in the Appendix). Hallucinated predictions, highlighted in red, consistently exhibit a low

---

[2]For further details on the first-order Taylor expansion, please refer to the Appendix.

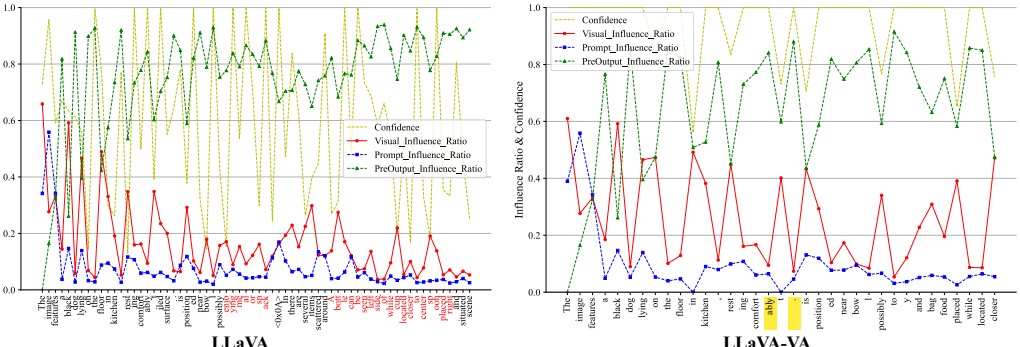

**LLaVA:** The image features a black dog lying on the floor in a kitchen, resting comfortably on a tiled surface. The dog is positioned near a bowl, possibly enjoying a meal or a snack. In the kitchen, there are several items scattered around. A bottle can be seen on the left side of the dog, while a cup is located closer to the center. A spoon is placed near the right side of the dog, and a bowl is situated on the right side of the scene.

**LLaVA-VA:** The image features a black dog lying on the floor of a kitchen, resting comfortably on a tile floor. The dog is positioned near a bowl, a toy, and a bag of cat food. The bowl is placed on the floor, while the toy is located closer to the dog.

Figure 2: Comparison of component influence ratios in image captioning with and without GACD-VA. (Left) Captions generated using LLaVA-v1.5 show that as sequence length increases, the influence of previous output tokens continues to rise, diverging from the influence of visual tokens and prompt tokens. Hallucinated predictions, highlighted in red, are characterized by a low visual influence. (Right) GACD-VA amplifies the influence of visual tokens, thereby increasing confidence and reducing hallucinations. Punctuation marks and suffixes exhibit a low visual influence ratio.

visual influence ratio. As more tokens are generated, the influence of both visual and prompt tokens diminishes, while the contribution of previous output tokens increases. This behavior aligns with findings from previous studies Zhou et al. (2023); Favero et al. (2024) and reflects the tendency of MLLMs to prioritize more recent elements in the sequence over earlier inputs. Additionally, the nature of the output token affects the degree of visual influence. For instance, punctuation marks (such as '.') or suffixes (such as 'ably' in 'comfortably') tend to have a lower visual influence ratio. This is intuitive, as these tokens rely more on linguistic context and are less dependent on visual information. This observation underscores the value of hallucination mitigation methods based on influence analysis, with GACD-VA (Sec. 4.1) offering more meaningful insights compared to approaches that focus solely on the length of the generated sequence.

**Co-occurrence Bias** occurs when models overly rely on associations between elements that frequently appear together in the training data. Fig. 3a illustrates an example of co-occurrence hallucination, where mPLUG-Owl2 with GACD-VA (mPLUGOwl-VA) incorrectly predicts 'dining table' in the presence of 'chair' in the image, demonstrating that merely amplifying visual influence is insufficient to mitigate such hallucinations. We analyze the influence of visual tokens on the predic-

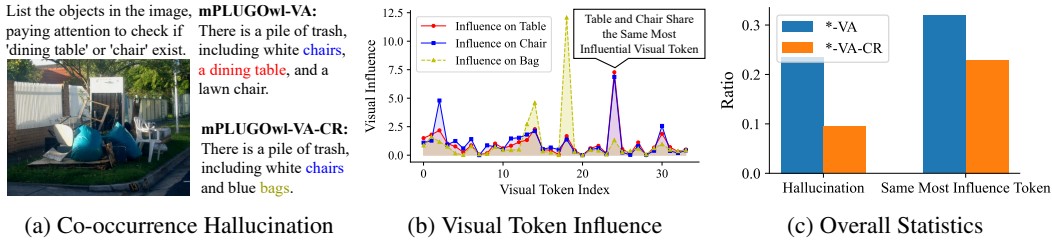

(a) Co-occurrence Hallucination     (b) Visual Token Influence     (c) Overall Statistics

List the objects in the image, paying attention to check if 'dining table' or 'chair' exist.

**mPLUGOwl-VA:** There is a pile of trash, including white chairs, a dining table, and a lawn chair.

**mPLUGOwl-VA-CR:** There is a pile of trash, including white chairs and blue bags.

Figure 3: (a) Co-occurrence hallucination of 'dining table' in the presence of a 'chair'. (b) Visualization of individual visual token influence for the left example shows that the token with the highest influence on 'chair' also has the highest influence on 'table'. (c) Summary statistics for 100 chair-only and 100 table-only images, including hallucination rate and the percentage of cases where both objects share the same most influential visual token. GACD-CR successfully reduces both metrics.

tions of the distracting object ('chair'), the hallucinated object ('table'), and unrelated correct object ('bag'). Fig. 3b shows individual visual token influences: $\left| \mathbf{g}^v_{m_c s} \right|_{y_{m_c} = chair}$, $\left| \mathbf{g}^v_{m_t s} \right|_{y_{m_t} = table}$, and

$\left|\mathbf{g}^v_{m_b s}\right|_{y_{m_b}=bag}$, revealing that the visual token most influential for the hallucinated 'table' also significantly contributes to 'chair', whereas the unrelated 'bag' is influenced by different visual tokens.

To further investigate, we collected 100 chair-only and 100 table-only images from the MSCOCO Lin et al. (2014) evaluation dataset. Experimental results, shown in the blue bar in Fig. 3c, reveal that in the presence of either a 'chair' or 'table' in the image, the hallucination rate for the other object is 23.5%, with a shared most influential visual token rate of 31.9% in these hallucinations, confirming our observation. This finding also inspired the development of GACD-CR (Sec. 4.2).

**Long-term bias** typically arises when generating long outputs, as hallucinations become more likely the further the output drifts from the visual context. This trend is evident in the LLaVA-v1.5 results shown in Fig. 2, where hallucinations like 'bottle', 'cup', and 'spoon' occur as the influence of visual tokens $r^v$ decreases with increasing $m$. Based on this, we also propose a stopping criterion.

## 4 SELF-REFLECTIVE CONTRASTIVE DECODING

Building on the bias analysis, we propose a novel approach, GACD, as outlined in Fig. 4, which uses self-reflection to achieve token influence awareness and adjust logits during decoding. This approach follows the principle of contrastive decoding, aiming to enlarge the Kullback-Leibler (KL) divergence [3] between the contrastive logits distribution, generated by a subset of input tokens, and the joint logits distribution, generated by all input tokens. This increase in KL divergence enhances the influence of tokens outside the subset. The contrastive decoding weight, $\alpha$, is calculated based on the influence of visual, prompt, and previous outputs on both the original and contrastive logits, ensuring that the adjusted logits receive a balanced visual influence. This calculation relies on a local linear assumption of MLLM functions. While not strictly precise, it offers significantly greater accuracy compared to using a fixed hyperparameter or relying solely on sequence length. Additionally, we propose a stopping criterion based on the visual influence ratio $r^v$. If the visual ratio for the sentence-starting token falls below a defined threshold, early stopping is triggered to halt further output generation.

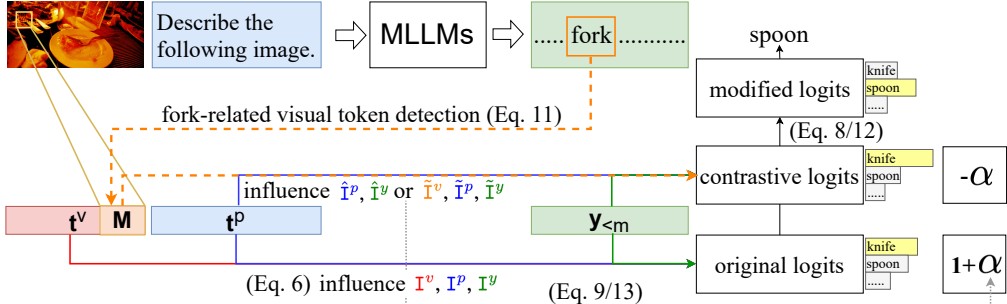

Figure 4: Overview of the proposed GACD method. It uses contrastive decoding to amplify the influence of essential visual tokens by generating contrastive logits from all input tokens except the essential visual ones. In GACD-VA (Sec. 4.1), contrastive logits are generated from prompt and previous output tokens to amplify all visual tokens. In GACD-CR (Sec. 4.2), visual tokens related to distracting objects are first detected, and then, along with prompt and previous output tokens, are used to generate contrastive logits, thereby amplifying the influence of non-distracting visual tokens.

### 4.1 VISUAL-TOKEN AMPLIFICATION

We aim to increase the influence of visual tokens, ensuring alignment with both the prompt and the previous output tokens. In VQA tasks, it is crucial for visual token influence to match the question prompt to ensure visually grounded responses. Likewise, in open-ended generation, maintaining visual token influence in balance with previous outputs is essential to prevent visual forgetting. To achieve this, our method adjusts the logits by incorporating contrastive logits generated from the prompt and previous output tokens:

$$\hat{\mathcal{F}}_{\theta^*}(\mathbf{t}^v, \mathbf{t}^p)_m = (1 + \alpha_m)\mathcal{F}_{\theta^*}(\mathbf{t}^v, \mathbf{t}^p)_m - \alpha_m \mathcal{F}_{\theta^*}(\mathbf{t}^p)_m, \tag{8}$$

---

[3]A detailed explanation of why contrastive decoding enlarges KL divergence is provided in the Appendix.

where $\alpha_m$ is calculated by considering the influence of each component and explicitly matching the dominant influence from either the prompt or previous outputs:

$$\alpha_m = \frac{\mathtt{I}^\star - \mathtt{I}^v}{\mathtt{I}^v + \hat{\mathtt{I}}^\star - \mathtt{I}^\star}, \quad \text{where } \star = \begin{cases} p & \text{if } \mathtt{I}^p \geq \mathtt{I}^y \\ y & \text{otherwise} \end{cases}, \tag{9}$$

where $\hat{\mathtt{I}}^p := \sum_{n=1}^{N} \left| \frac{\partial (\mathcal{F}_{\theta^*}(\mathbf{t}^p))_m}{\partial t_n^p} \right|$ and $\hat{\mathtt{I}}^y := \sum_{i=1}^{m-1} \left| \frac{\partial (\mathcal{F}_{\theta^*}(\mathbf{t}^p))_m}{\partial y_i} \right|$ represent the influence of the prompt and previous output, respectively, on the contrastive logits.

Furthermore, unlike existing contrastive decoding methods Zhou et al. (2023); Favero et al. (2024); Leng et al. (2023), which rely on adaptive plausibility constraints such as the prediction confidence and require experiments to determine the optimal threshold, our approach operates as long as the prediction confidence is below $100\%$. Instead, our method explicitly imposes a constraint to prevent the influence of the prompt from becoming negative:

$$\alpha_m \leq \frac{\mathtt{I}^p}{\hat{\mathtt{I}}^p - \mathtt{I}^p}, \quad \text{if } \hat{\mathtt{I}}^p > \mathtt{I}^p. \tag{10}$$

## 4.2 CO-OCCURRENCE REDUCTION

To reduce co-occurrence hallucination, we first identify visual tokens related to distracting objects by analyzing those with the highest influence, then mitigate their impact in subsequent predictions. A mask is applied to flag these distracting-object-related visual tokens, focusing on object (noun) output tokens. When a noun is predicted, the corresponding visual token with the highest influence is flagged. The mask $\mathcal{M}_{ms}^v$ aggregates visual tokens related to all previously mentioned objects:

$$\mathcal{M}_{ms}^v = \mathcal{M}_{1s}^v \vee \mathcal{M}_{2s}^v \vee \cdots \vee \mathcal{M}_{(m-1)s}^v, \quad \mathcal{M}_{is}^v = \begin{cases} 1 & \text{if } \left| \mathbf{g}_{is}^{vc} \right| \geq \left| \mathbf{g}_{i(1...S)}^{vc} \right|, y_i \text{ is noun} \\ 0 & \text{otherwise} \end{cases}, \tag{11}$$

where $c$ in $\mathbf{g}_{is}^{vc}$ indicates that the gradient is specifically from the predicted object class, rather than from the overall predicted logits across the entire vocabulary classes.

During prediction, if the output is a noun, contrastive logits are generated from both text and distracting-object-related visual tokens, modifying equation 8 as follows:

$$\tilde{\mathcal{F}}_{\theta^*}(\mathbf{t}^v, \mathbf{t}^p)_m = (1 + \tilde{\alpha}_m)\mathcal{F}_{\theta^*}(\mathbf{t}^v, \mathbf{t}^p)_m - \tilde{\alpha}_m \mathcal{F}_{\theta^*}(\mathbf{t}^v \times \mathcal{M}_m^v, \mathbf{t}^p)_m, \tag{12}$$

where $\times \mathcal{M}^v$ indicates the mask selection, and $\tilde{\alpha}_m$ controls the KL divergence between logits conditioned on only distracting-object-related visual tokens and those conditioned on all visual tokens. We modify equation 9 as follows:

$$\tilde{\alpha}_m = \frac{\mathtt{I}^\star - \mathtt{I}^v}{\mathtt{I}^v - \tilde{\mathtt{I}}^v + \tilde{\mathtt{I}}^\star - \mathtt{I}^\star}, \quad \star = \begin{cases} p & \text{if } \mathtt{I}^p \geq \mathtt{I}^y \\ y & \text{otherwise} \end{cases}, \tag{13}$$

where $\tilde{\mathtt{I}}^v := \sum_{s=1}^{S} \left| \frac{\partial (\mathcal{F}_{\theta^*}(\mathbf{t}^v \times \mathcal{M}_{ms}^v, \mathbf{t}^p))_m}{\partial t_s^v} \right|$, $\tilde{\mathtt{I}}^p := \sum_{n=1}^{N} \left| \frac{\partial (\mathcal{F}_{\theta^*}(\mathbf{t}^v \times \mathcal{M}_{ms}^v, \mathbf{t}^p))_m}{\partial t_n^p} \right|$ and $\tilde{\mathtt{I}}^y := \sum_{i=1}^{m-1} \left| \frac{\partial (\mathcal{F}_{\theta^*}(\mathbf{t}^v \times \mathcal{M}_{ms}^v, \mathbf{t}^p))_m}{\partial y_i} \right|$ represent the influence of visual, prompt, and previous output tokens, respectively, on the contrastive logits.

Since distracting-object-related visual tokens are included to generate the contrastive logits, alongside prompt tokens, we ensure that the influence of these visual tokens does not become negative like that of the prompt tokens. Accordingly, we modify the constraint in equation 10 as follows:

$$\tilde{\alpha}_m \leq \min \left( \frac{\mathtt{I}^d}{\tilde{\mathtt{I}}^v - \mathtt{I}^d}, \frac{\mathtt{I}^p}{\tilde{\mathtt{I}}^p - \mathtt{I}^p} \right), \quad \text{if } \tilde{\mathtt{I}}^v > \mathtt{I}^d, \quad \tilde{\mathtt{I}}^p > \mathtt{I}^p, \tag{14}$$

where $\mathtt{I}^d = \sum_{s=1}^{S} \left| \mathbf{g}_{ms}^v \times \mathcal{M}_{ms}^v \right|$ represents the influence of distracting-object-related visual tokens.

## 5 EXPERIMENTS

We evaluate our method across various MLLMs and datasets, comparing it with SOTA hallucination mitigation methods. If not specified otherwise, the experiments default to using greedy sampling.

## 5.1 EXPERIMENTAL SETTING

**Models**. We apply and evaluate our methods on widely-used models including LLaVA v1.5-7b, LlaVA v1.6-7b Liu et al. (2024a), InstructBLIP Dai et al. (2023), and mPLUG-Owl2 Ye et al. (2024).

**Implementation Details**. The maximum output length is set to 256 across all models, with other model parameters kept at their defaults. To avoid excessive modification, we limit $\alpha$ less than 5 for discriminative tasks and 3 for generative tasks. The early stopping thresholds used in the experiments are as follows: LLaVA-v1.5 and LLaVA-v1.6: 7%, InstrucBLIP: 25%, and mPLUG-Owl2: 2.5%. All experiments are performed on an NVIDIA A40 GPU with batch size of 1.

**Datasets and Evaluation Metrics**. In alignment with established evaluation standards from previous studies Zhao et al. (2024); Yin et al. (2023); Zhou et al. (2023); Leng et al. (2023), we evaluate our hallucination elimination method on discriminative benchmarks, including POPE Li et al. (2023) and the discriminative component of Amber Wang et al. (2023), as well as on open-ended benchmarks, such as the VQA benchmark LLaVa-QA90 Liu et al. (2024b) and image captioning benchmarks from MSCOCO Lin et al. (2014) and Amber Wang et al. (2023). For discriminative tasks, hallucination manifests as a misclassification problem $p(\mathbf{y} = \mathbf{yes}|\mathcal{F}_\theta, \mathbf{t}^p \notin \mathbf{t}^v, \mathbf{t}^p)$, we report both accuracy and F1 score. For open-ended VQA, GPT-4V OpenAI et al. (2024) is used to score both accuracy (Acc) and detailedness (Det) on a scale of 10. For image captioning, we focus on object hallucination and report the Caption Hallucination Assessment with Image Relevance (CHAIR) Rohrbach et al. (2018) score, which includes sentence-level ($C_S$) and instance-level ($C_I$) percentages, instance-level recall $R$, and the average generated length ($Len$).

## 5.2 EXPERIMENTAL RESULTS

**Results on POPE**. Tab. 1 and Tab.4 in the Appendix show that our methods significantly enhance baseline MLLMs, and outperform SOTA techniques, including the hallucination correction method Woodpecker Yin et al. (2023), and contrastive decoding approaches like VCD Leng et al. (2023), M3ID Favero et al. (2024) and AVISC Woo et al. (2024). These results suggest that explicitly and precisely balancing key biases provides substantial benefits. Our improvements are achieved solely through self-reflection and the application of GACD-VA, as GACD-CR and early stopping are not applicable due to the one-word output constraint of the discriminative task. Notably, these improvements are more pronounced on the mPLUG-Owl2 baseline compared to InstructBLIP and LLaVA-v1.5. This disparity may be because InstructBLIP already exhibits a relatively high visual influence ratio, as shown in Fig. 5a. In contrast, LLaVA-v1.5 tends to be overconfident about object existence, having 100% confidence even when the visual influence ratio is less than 30%, as illustrated in Fig. 5b. On the other hand, mPLUG-Owl2 is less overconfident and has a lower visual influence, which likely explains why it shows the greatest improvement.

Table 1: Discriminative VQA Comparison on POPE Li et al. (2023) in MSCOCO Adversarial Setting; Other Results from Zhao et al. (2024), Woo et al. (2024).

| Method | LLaVA-v1.5 | | InstructBLIP | | mPLUG-Owl2 | |
|---|---|---|---|---|---|---|
| | Acc ↑ | F1 ↑ | Acc ↑ | F1 ↑ | Acc ↑ | F1 ↑ |
| Original | 79.0 | 81.1 | 71.6 | 74.7 | 71.5 | 76.6 |
| Greedy | 79.4 | 81.6 | 79.8 | 81.4 | 72.5 | 77.5 |
| M3ID | 77.7 | 79.7 | 76.0 | 77.8 | - | - |
| AVISC | 77.5 | 79.6 | 81.6 | 81.9 | - | - |
| Woodpecker | 80.5 | 80.6 | 79.0 | 78.6 | 77.5 | 76.9 |
| VCD | 80.9 | 81.3 | 79.6 | 79.5 | - | - |
| Ours | **83.5** | **82.1** | **82.5** | **82.1** | **84.2** | **83.7** |

**Results on LLaVA-QA90**. We further evaluate our full method, in open-ended VQA tasks on the LLaVA-QA90 Liu et al. (2024b) dataset. The results in Tab. 2 demonstrate the superiority of our approach, consistently improving baseline accuracy and detailness, with at least a 1.44 accuracy improvement and a 0.7 detailness improvement. As illustrated in Fig. 2, in open-ended VQA, our method explicitly amplifies the visual influence to align with the question prompt at the beginning, ensuring that responses are visually grounded. Additionally, it matches the influence of previous outputs as the output length increases, preventing visual forgetting. This explicit balance allows our approach to surpass the VCD method Leng et al. (2023), which also employs contrastive decoding.

Table 2: Open-ended VQA Comparison on LLaVA-QA90; Other Results from Leng et al. (2023).

| Method | LLaVA-v1.5 | | InstructBLIP | | mPLUG-Owl2 | |
|---|---|---|---|---|---|---|
| | Acc ↑ | Det ↑ | Acc ↑ | Det ↑ | Acc ↑ | Det ↑ |
| Baseline | 3.23 | 3.54 | 3.84 | 4.07 | 4.07 | 4.33 |
| VCD | 4.15 | 3.85 | 4.23 | 4.69 | - | - |
| Ours | **6.20** | **5.13** | **6.28** | **4.77** | **6.69** | **6.28** |

**CHAIR Evaluations on MSCOCO**. For the image captioning task, we evaluate our approach on a subset of MSCOCO, following the evaluation settings in Deng et al. (2024). As shown in Tab. 3, our approach consistently and significantly reduces hallucination rates at both the object level ($C_S$) and image level ($C_I$), with particularly strong reductions at the image level ($C_I$). This demonstrates its effectiveness in mitigating all types of hallucinations. Hallucination rates are strongly correlated with caption length, where increased length often leads to higher hallucination risks. Compared to SOTA methods, our approach more effectively reduces hallucination rates while maintaining similar caption lengths, demonstrating robustness in balancing detail retention with accuracy. It ensures precise and reliable image captions without compromising between length and hallucination risk.

Table 3: Image Captioning Comparison on MSCOCO subset. Other Results from Deng et al. (2024)

| Method | LLaVA-v1.5 | | | InstructBLIP | | | mPLUG-Owl2 | | |
|---|---|---|---|---|---|---|---|---|---|
| | $C_S$ ↓ | $C_I$ ↓ | Len ↑ | $C_S$ ↓ | $C_I$ ↓ | Len ↑ | $C_S$ ↓ | $C_I$ ↓ | Len ↑ |
| Baseline | 48.8 | 13.4 | **99.8** | 57.8 | 16.5 | **101.3** | 59.2 | 17.6 | **105.3** |
| OPERA | 49.5 | 13.7 | 85.7 | 51.5 | 15.6 | 85.8 | 48.5 | 16.1 | 86.1 |
| VCD | 44.6 | 12.5 | 85.7 | 63.2 | 19.5 | 92.5 | 51.4 | 16.0 | 89.6 |
| Ours | **41.0** | **10.9** | 85.0 | **47.4** | **13.4** | 93.9 | **45.0** | **12.4** | 83.5 |

**Results on Amber**. Evaluation on the Amber dataset focuses on both image captioning and comprehensive discriminative performance, covering not only object existence but also categories like attributes and relationships. Tab. 4 and Fig. 6 demonstrate that our method consistently improves the performance of various MLLMs across both tasks. Notably, LLaVA-v1.5 and mPLUG-Owl2 exhibit significant improvements compared to InstructBLIP and LLaVA-v1.6. Our method even enhances LLaVA-v1.5's overall score to outperform LLaVA-v1.6, likely due to differences in the models' original visual influence ratios, as InstructBLIP and LLaVA-v1.6 already have more balanced visual contributions. Additionally, LLaVA-v1.6 emphasizes recall over precision, as indicated by its high scores in both coverage (cov) and recall (R). In terms of discriminative performance, our method achieves a better balance between precision and recall, resulting in improved F1 scores. Furthermore, Fig. 6 shows that our method enhances performance across all categories, with particularly significant gains in existence, attributes, and state. This can be attributed to the increased influence of visual tokens, benefiting categories that are easily discernible from the visual inputs.

Table 4: Results on the AMBER Dataset. Results marked with ⋆ are reported in Wang et al. (2023).

| Method | w/Ours | Generative Task | | | | Discriminative Task | | | | Score↑ |
|---|---|---|---|---|---|---|---|---|---|---|
| | | cha ↓ | cov ↑ | hal ↓ | cog ↓ | acc ↑ | P ↑ | R ↑ | F1 ↑ | |
| LLaVA-v1.5 | ✗⋆ | 7.8 | 51.0 | 36.4 | 4.2 | 72.0 | **93.2** | 62.4 | 74.7 | 83.5 |
| | ✓ | **6.6** | **54.7** | **33.0** | **2.9** | **80.3** | 82.9 | **89.3** | **86.0** | **89.7** |
| LLaVA-v1.6 | ✗ | 9.9 | 56.7 | 47.4 | 4.3 | 80.3 | 82.9 | **89.3** | 86.0 | 88.5 |
| | ✓ | **8.7** | **58.3** | **43.8** | **2.5** | **81.2** | **85.2** | 88.8 | **87.0** | **89.2** |
| InstructBLIP | ✗⋆ | 8.8 | **52.2** | 38.2 | 4.4 | 76.5 | 84.5 | **79.0** | 81.7 | 86.5 |
| | ✓ | **7.2** | 52.1 | **34.3** | **3.6** | **78.1** | **88.8** | 76.6 | **82.2** | **87.5** |
| mPLUG-Owl2 | ✗⋆ | 10.6 | 52.0 | 39.9 | 4.5 | 75.6 | **95.0** | 66.9 | 78.5 | 84.0 |
| | ✓ | **7.5** | **53.6** | **34.7** | **4.0** | **82.1** | 87.0 | **86.2** | **86.6** | **89.6** |

**Component analysis** in Tab. 5 shows each proposed component contributes to the overall performance. GACD-VA significantly reduces hallucinations while improving object recall. GACD-CR further mitigates co-occurrence bias, a residual form of the text-visual bias addressed by GACD-VA, leading to additional hallucination reduction. The greater reduction at the image level ($C_I$) compared to the object level ($C_S$) confirms that addressing this residual bias leads to a more comprehensive

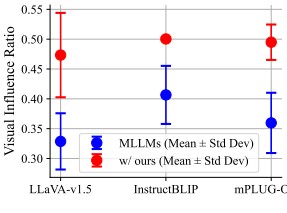
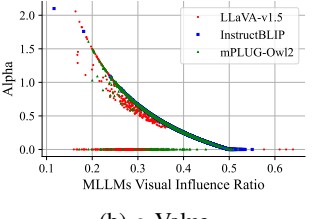
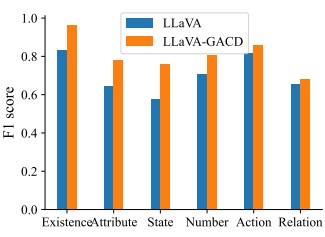

(a) Visual Influence Ratio      (b) $\alpha$ Value

Figure 5: (a) Visual influence ratios across the POPE dataset: the ratios are less than $50\%$ with original MLLMs, GACD successfully increases them to nearly $50\%$. (b) $\alpha$ value visualization: $\alpha$ inversely correlates with the visual influence ratio, as expected. It drops to 0 when the ratio exceeds $50\%$, indicating no further adjustments are needed, and also when the ratio is below $50\%$ in cases of $100\%$ prediction confidence.

Figure 6: F1 scores for subcategories of the Amber discriminative task using LLaVA-v1.5. GACD consistently improves performance, with the largest gain in 'State' and the smallest in 'Relation'.

elimination of all types of hallucinations, improving overall image accuracy. The early-stop mechanism effectively reduces hallucinations, with only a minor trade-off in object recall.

Fig. 5a shows that GACD-VA successfully increases the overall visual influence to match that of the object-existent question prompt in the POPE dataset [4]. The visualization of $\alpha$ in Fig. 5b indicates that the $\alpha$ range of LLaVA-v1.5, InstructBlip, and mPLUG-Owl2 is primarily between 0 and 1.5. Furthermore, LLaVA-v1.5 and mPLUG-Owl2 tend to exhibit overconfidence regarding object existence in VQA tasks, as $\alpha$ is set to zero even when the visual influence ratio is below $50\%$. LLaVA-v1.5 is more prone to overconfidence, with many samples falling within a visual influence range of $15\%$ to $35\%$, compared to mPLUG-Owl2's range of $20\%$ to $45\%$. Fig. 2 and Fig.1 in the Appendix show that GACD-VA effectively increases the influence of visual tokens, aligning them with the prompt and previous outputs in open-ended generation tasks. This results in higher prediction confidence and a reduction in hallucinations. Fig. 3c illustrates the effectiveness of GACD-CR, reducing the hallucination rate of predicting both 'table' or 'chair' in single-object images.

Table 5: Component Analysis Using the CHAIR Metric

| Components | | | LLaVA-v1.5 | | | | InstructBLIP | | | | mPLUG-Owl2 | | | |
|---|---|---|---|---|---|---|---|---|---|---|---|---|---|---|
| VA | CR | ES | $C_S \downarrow$ | $C_I \downarrow$ | $R \uparrow$ | $Len \uparrow$ | $C_S \downarrow$ | $C_I \downarrow$ | $R \uparrow$ | $Len \uparrow$ | $C_S \downarrow$ | $C_I \downarrow$ | $R \uparrow$ | $Len \uparrow$ |
| | | | 48.8 | 13.4 | 78.6 | **99.8** | 57.8 | 16.5 | 73.6 | 101.3 | 59.2 | 17.6 | 75.8 | **105.3** |
| ✓ | | | 46.4 | 11.6 | 79.0 | 95.6 | 53.6 | 15.1 | **75.3** | **108.4** | 52.6 | 14.4 | **78.2** | 95.6 |
| ✓ | ✓ | | 46.2 | 11.3 | **79.4** | 95.5 | 53.2 | 14.0 | 74.6 | 105.7 | 52.3 | 14.2 | 78.0 | 95.5 |
| ✓ | ✓ | ✓ | **41.0** | **10.9** | 77.3 | 85.0 | **47.4** | **13.4** | 72.3 | 93.9 | **45.0** | **12.4** | 74.9 | 83.5 |

## 6 CONCLUSION

In this work, we address the challenge of hallucinations in MLLMs by targeting three key biases: text-visual, co-occurrence, and long-term biases. We introduce the GACD framework, which uses gradient-based self-reflection to jointly analyze and balance these biases, mitigating hallucinations without requiring costly resources or tuning. The proposed GACD is the first method capable of fully addressing co-occurrence hallucinations, even when distracting elements are present in the visual input. It amplifies essential visual influences to balance text-visual and co-occurrence biases, and introduces a stopping criterion to mitigate long-term hallucinations.

Our method is limited to white-box MLLMs, as it relies on access to gradients. Its effectiveness depends on the importance and clarity of visual information relative to the prompt. Questions about object existence are straightforward and primarily rely on visual inputs, whereas questions about relationships are less direct and require inference beyond visual inputs. As a post-processing technique, our method does not involve model training. In future work, we aim to explore how insights from GACD can guide and improve training strategies for enhanced visual perception in MLLMs.

---

[4]Note that the standard deviation of the visual ratio for InstructBlip, after applying GACD-VA, is small. This occurs because InstructBlip is not overly confident, enabling our method to adjust the visual ratio to $50\%$ for nearly all samples.

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
