# Mitigating Multimodal Hallucinations via Gradient-based Self-Reflection

## A  Appendix

### A.1  First Order Taylor Expansion

The first order Taylor expansion of $\mathcal{F}_{\theta^*}$ w.r.t $\mathbf{t}^v, \mathbf{t}^p$ and, all the previous output tokens $\mathbf{y}_i$ with $i < m$:

$$
\begin{aligned}
\mathcal{F}_{\theta^*}(\mathbf{t}^v, \mathbf{t}^p)_m \approx & \mathcal{F}_{\theta^*}(\mathbf{t}^{v(0)}, \mathbf{t}^{p(0)})_m + \sum_{s=1}^{S} \left.\frac{\partial(\mathcal{F}_{\theta^*})_m}{\partial t_s^v}\right|_{\mathbf{t}^{v(0)}} (t_s^v - t_s^{v(0)}) \\
& + \sum_{n=1}^{N} \left.\frac{\partial(\mathcal{F}_{\theta^*})_m}{\partial t_n^p}\right|_{\mathbf{t}^{p(0)}} (t_n^p - t_n^{p(0)}) + \sum_{i=1}^{m-1} \left.\frac{\partial(\mathcal{F}_{\theta^*})_m}{\partial y_i}\right|_{y_i^{(0)}} (y_i - y_i^{(0)}) \quad (1) \\
= & \sum_{s=1}^{S} \mathbf{g}_{ms}^v \cdot t_s^v + \sum_{n=1}^{N} \mathbf{g}_{mn}^p \cdot t_n^p + \sum_{i=1}^{m-1} \mathbf{g}_{mi}^y \cdot y_i + Const,
\end{aligned}
$$

where $y_i^{(0)} := \mathcal{F}_{\theta^*}(\mathbf{t}^{(0)}, \mathbf{t}^{p(0)})_i$ is the $i^{\text{th}}$ output token given the input.

### A.2  Interpreting Contrastive Decoding through KL Divergence

Kullback-Leibler (KL) divergence can be used to interpret contrastive decoding, For example, in GACD-VA, it measures the divergence between the reference distribution $p(y_{cm}|\mathcal{F}_{\theta^\star}, \mathbf{t}^p, y_{<m})$ to the visual joint distribution $p(y_{cm}|\mathcal{F}_{\theta^\star}, \mathbf{t}^v, \mathbf{t}^p, y_{<m})$.

$$
\begin{aligned}
\mathrm{D}_{KL} = & \sum_c p(y_{cm}|\mathcal{F}_{\theta^\star}, \mathbf{t}^v, \mathbf{t}^p, y_{<m})\log\big(\frac{p(y_{cm}|\mathcal{F}_{\theta^\star}, \mathbf{t}^v, \mathbf{t}^p, y_{<m})}{p(y_{cm}|\mathcal{F}_{\theta^\star}, \mathbf{t}^p, y_{<m})}\big) \\
= & \sum_c p(y_{cm}|\mathcal{F}_{\theta^\star}, \mathbf{t}^v, \mathbf{t}^p, y_{<m})(\log(p(y_{cm}|\mathcal{F}_{\theta^\star}, \mathbf{t}^v, \mathbf{t}^p, y_{<m})) - \log(p(y_{cm}|\mathcal{F}_{\theta^\star}, \mathbf{t}^p, y_{<m}))) \\
= & \sum_c p(y_{cm}|\mathcal{F}_{\theta^\star}, \mathbf{t}^v, \mathbf{t}^p, y_{<m})(\mathcal{F}_{\theta^*}(\mathbf{t}^v, \mathbf{t}^p)_m[c] - \log(\sum \exp(\mathcal{F}_{\theta^*}(\mathbf{t}^v, \mathbf{t}^p)_m)) \\
& - \mathcal{F}_{\theta^*}(\mathbf{t}^p)_m[c] + \log(\sum \exp(\mathcal{F}_{\theta^*}(\mathbf{t}^p)_m))) \\
= & \sum_c p(y_{cm}|\mathcal{F}_{\theta^\star}, \mathbf{t}^v, \mathbf{t}^p, y_{<m})(\underline{(\mathcal{F}_{\theta^*}(\mathbf{t}^v, \mathbf{t}^p)_m - \mathcal{F}_{\theta^*}(\mathbf{t}^p)_m)}[c] + Const),
\end{aligned}
$$
$$(2)$$

where $p(y_{cm}|\mathcal{F}_{\theta^\star}, \mathbf{t}^v, \mathbf{t}^p, y_{<m}) = \sigma(\mathcal{F}_{\theta^*}(\mathbf{t}^v, \mathbf{t}^p)_m)$, $p(y_{cm}|\mathcal{F}_{\theta^\star}, \mathbf{t}^p, y_{<m}) = \sigma(\mathcal{F}_{\theta^*}(\mathbf{t}^p)_m)$ and $c$ represents a class in the predefined vocabulary. The adjustment term $\mathcal{F}_{\theta^*}(\mathbf{t}^v, \mathbf{t}^p)_m - \mathcal{F}_{\theta^*}(\mathbf{t}^p)_m$ increases the KL divergence, thereby emphasizing the impact of visual tokens.

### A.3  Different Sampling Strategies

Tab. 1 presents an ablation study on sampling strategies, showing that our method performs better with greedy sampling in both discriminative and generative VQA tasks. This reflects the precision of our logit adjustments, where the highest-confidence prediction is generally the most accurate.

Table 1: Ablation Study on Sampling Strategies.

| strategy | POPE MSCOCO Adversarial | | | | | | LLaVA-QA90 | | | | | |
| | LLaVA-v1.5 | | InstructBLIP | | mPLUG-Owl2 | | LLaVA-v1.5 | | InstructBLIP | | mPLUG-Owl2 | |
| | Acc | F1 | Acc | F1 | Acc | F1 | Acc | Det | Acc | Det | Acc | Det |
|---|---|---|---|---|---|---|---|---|---|---|---|---|
| random | 82.3 | 81.1 | 82.2 | 81.8 | 83.2 | 82.9 | 5.79 | 4.74 | 5.98 | 4.64 | 6.07 | 5.72 |
| greedy | 83.5 | 82.1 | 82.5 | 82.1 | 84.2 | 83.7 | 6.20 | 5.13 | 6.28 | 4.77 | 6.69 | 6.28 |

## A.4    ABLATION STUDY ON THE GACD PROCESS WITH 100% CONFIDENCE

We conducted an ablation study to evaluate the effect of processing GACD even with 100% confidence. Results in Tab. 2 show that applying GACD with 100% confidence yields only marginal improvements, indicating that the benefits of GACD are more pronounced when the model's confidence is lower.

Table 2: Ablation Study on 100% Confidence with mPLUG-Owl2

| w/ 100% Confidence | POPE MSCOCO Adversarial | | LLaVA-QA90 | |
| | Acc | F1 | Acc | Det |
|---|---|---|---|---|
| ✗ | 84.2 | 83.7 | 6.69 | 6.28 |
| ✓ | 84.2 | 83.7 | 6.75 | 6.29 |

## A.5    MLLMS ARCHITECTURES

Tab. 3 shows detailed information about the vision encoder and LLM components of the MLLM architectures used in our experiments.

Table 3: Details of the used MLLM architectures.

| MLLMs | Vision encoder | LLM |
|---|---|---|
| LLaVA-v1.5 | CLIP-L-336px | Vicuna-v1.5-7B |
| LLaVA-v1.6 | CLIP-L-336px | Vicuna-v1.5-7B |
| InstructBLIP | BLIP-2 | Vicuna-v1.1-7B |
| mPLUG-Owl2 | CLIP-L | LLaMA-2-7B |

## A.6    OTHER RESULTS OF POPE

We report our experimental results on the POPE dataset, in addition to MSCOCO and adversarial settings, in Tab. 4. The results indicate that our method improves performance across all baseline MLLMs, with more significant gains observed in the adversarial setting. This is expected, as visual inputs provide crucial clues for identifying adversarial objects.

## A.7    INFLUENCE RATIO IN VQA

Fig. 1 illustrates the influence ratio across predicted tokens in VQA tasks, comparing baseline predictions with those obtained after applying the GACD-VA addon. The analysis confirms that as more tokens are generated, the influence of both visual and prompt tokens diminishes, while the contribution of previous output tokens steadily increases, shifting the primary drivers of the model's predictions. It also shows that visual tokens generally have a lower influence compared to prompt tokens across MLLMs, particularly at the beginning. The application of GACD-VA rebalances these influences, boosting the visual token influence to align with the dominant components—prompt at the beginning and previous outputs later—thereby reducing the likelihood of hallucinations in the model's predictions.

Table 4: More Results on POPE **?**.

| Dataset | Setting | w/Ours | LLaVA-v1.5 | | InstructBLIP | | mPLUG-Owl2 | |
|---------|---------|--------|------------|------|--------------|------|------------|------|
| | | | Acc ↑ | F1 ↑ | Acc ↑ | F1 ↑ | Acc ↑ | F1 ↑ |
| MSCOCO | Random | ✗ | 86.5 | 84.8 | 87.1 | 85.7 | 86.0 | 84.4 |
| | | ✓ | **86.8** | **85.1** | **87.9** | **86.8** | **87.9** | **87.1** |
| | Popular | ✗ | 85.5 | 83.8 | 84.2 | 83.6 | 84.6 | 83.2 |
| | | ✓ | **85.6** | **84.0** | **85.0** | **84.3** | **86.4** | **85.7** |
| A-OKVQA | Random | ✗ | 88.0 | **87.6** | 88.5 | 88.5 | 86.5 | 85.7 |
| | | ✓ | **88.1** | 87.4 | **88.8** | **88.8** | **88.4** | **88.1** |
| | Popular | ✗ | **85.5** | **85.1** | 81.9 | 83.1 | 82.4 | 82.2 |
| | | ✓ | **85.5** | **85.1** | **82.3** | **83.4** | **85.1** | **85.3** |
| | Adversarial | ✗ | 79.1 | 79.9 | 74.8 | 77.9 | 74.7 | 76.9 |
| | | ✓ | **79.5** | **80.1** | **75.3** | **78.2** | **78.2** | **79.9** |
| GQA | Random | ✗ | **88.9** | **88.2** | **87.2** | 87.1 | 85.2 | 84.0 |
| | | ✓ | **88.9** | **88.2** | **87.2** | **87.2** | **86.1** | **85.0** |
| | Popular | ✗ | 84.1 | **84.1** | 78.6 | **80.4** | 78.7 | 78.5 |
| | | ✓ | **84.2** | **84.1** | **78.8** | **80.4** | **81.0** | **80.5** |
| | Adversarial | ✗ | 80.8 | 81.3 | 75.9 | 78.4 | 76.4 | 76.8 |
| | | ✓ | **81.1** | **81.6** | **76.1** | **78.5** | **79.2** | **79.1** |

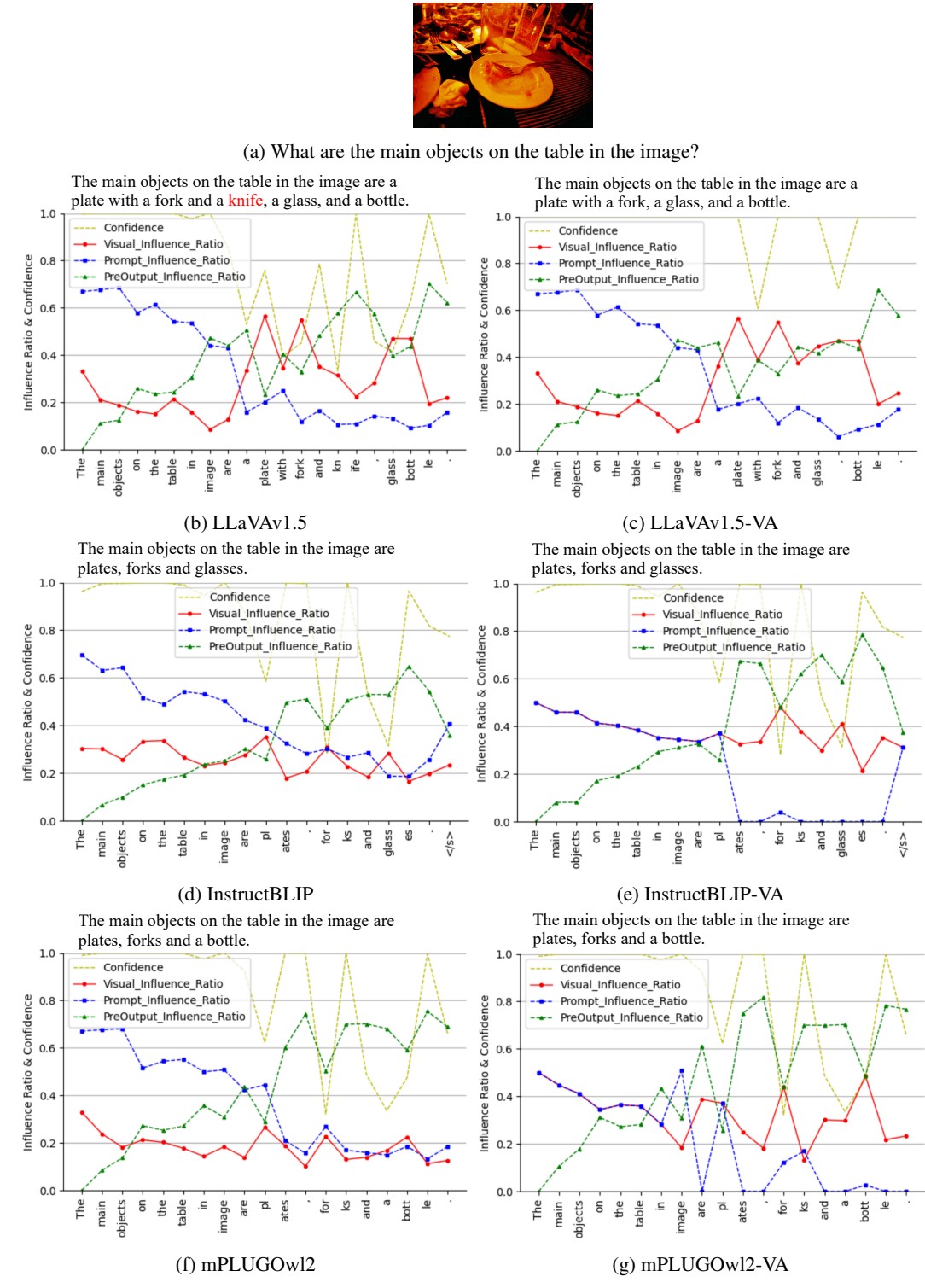

Figure 1: Influence Ratio across Predicted Tokens in VQA: (left) Baseline predictions; (right) Predictions with GACD-VA. In general, visual tokens initially contribute less than prompt tokens. As more tokens are generated, the contribution of visual tokens and prompt tokens decreases, while the influence of previous output tokens increases. GACD-VA effectively boosts the influence of visual tokens to match the dominant components—prompt tokens at the start and previous output tokens toward the end—thereby mitigating hallucinations.