# OpenReview forum: "Mitigating Multimodal Hallucinations via Gradient-based Self-Reflection"
_ICLR.cc/2025/Conference — ICLR 2025 Conference Withdrawn Submission_

### Official Review · Reviewer_CFEW · 2024-10-23

**Soundness:** 3
**Presentation:** 3
**Contribution:** 3
**Rating:** 3
**Confidence:** 4

**Summary:**

This paper explores the issue of object hallucination in large vision-language models (VLMs) and attributes the phenomenon to three key biases: text-visual bias, co-occurrence bias, and long-term bias. The authors propose a method to quantify the influence of each of these biases by calculating the gradient-based impact of input tokens on the output. Furthermore, they introduce the GACD approach, which balances the influence of hallucination-inducing tokens or applies early stopping to mitigate hallucinations.

**Strengths:**

### S1. The effort to categorize hallucination into text-visual bias, co-occurrence bias, and long-term bias is commendable. The classification is convincing, particularly as illustrated in Figure 1.

### S2. The idea of quantifying the influence of each bias is interesting and seems useful. Visualizing the impact of these biases in Figure 2 and Appendix Figure 1 was clear and helpful for understanding the distinctions between them.

**Weaknesses:**

### W1. The experimental validation is somewhat weak. The POPE results in Table 4 of the Appendix show only minor improvements in the Random and Popular settings, and there is no comparison with other methods such as VCD, AvisC, etc., which makes it difficult to assess the effectiveness of the proposed method comprehensively.

### W2. The paper lacks sufficient details regarding the GPT-4V prompt used in the open-ended QA evaluation. Providing a more detailed explanation would improve clarity.

### W3. Regarding hyperparameters, the choice of alpha values (set to less than 5 for discriminative tasks and 3 for generative tasks) lacks justification. A rationale for these values should be provided.

### W4. Early stopping criteria are determined differently for each model, but there is no explanation or ablation study to justify these choices. The lack of grounding for these decisions weakens the methodological rigor, and this applies to both points 3 and 4.

### W5. The paper claims that GACD is the first approach to fully address co-occurrence bias without extra resources or tuning. However, this claim seems questionable. Since co-occurrence bias and language prior often share the same underlying mechanisms, existing contrastive decoding methods like VCD, M3ID, and AvisC seem to address this issue inherently as well. Clarification is needed here.

**Questions:**

### Q1. Is there a specific reason why the POPE Adversarial setting in Table 1 performs significantly better?

### Q2. Why were experiments conducted only on the 7B model? How would the proposed method scale with models larger than 13B or smaller models? It would be helpful to understand performance variation across different model scales.

----

### Minor comment: Please use citep instead of cite, and reserve cite for the beginning of a sentence.

---

### Official Review · Reviewer_NgbB · 2024-11-01

**Soundness:** 2
**Presentation:** 3
**Contribution:** 2
**Rating:** 5
**Confidence:** 4

**Summary:**

The paper titled "Mitigating Multimodal Hallucinations via Gradient-Based Self-Reflection" presents a novel approach to address the issue of hallucination in MLLMs. The authors identify three primary biases causing hallucinations: text-visual bias, co-occurrence bias, and long-term bias. They propose a method called Gradient-based Influence-Aware Contrastive Decoding (GACD), which leverages gradient-based self-reflection to balance these biases and mitigate hallucinations without additional resources or tuning. The paper claims that GACD is the first method capable of fully addressing co-occurrence bias and provides extensive experimental results demonstrating its effectiveness in improving MLLM performance.

**Strengths:**

- The paper introduces a unique method, GACD, which uses gradient norms to quantify 'token influence' and self-reflection to balance biases, offering a new perspective on hallucination mitigation in MLLMs.
- GACD is the first approach that can fully address co-occurrence bias, which is a significant contribution to the field.
- The token influence analysis allows for a detailed understanding of biases at the individual sample level, which is a step forward from previous methods that relied on overall statistics.

**Weaknesses:**

- The paper's validation is based on relatively older versions of MLLMs, which may not represent the current state-of-the-art in the field. The effectiveness of GACD should be tested on the latest MLLMs, such as InternVL2 [1] and Qwen2-VL [2], to ensure that the findings are relevant and applicable to current research and industry standards.
- While the authors claim that their method is effective against all three types of hallucinations, the experimental section lacks a focused validation on each specific type of hallucination. A more detailed analysis targeting each bias individually would strengthen the paper's claims.
- The paper relies on benchmarks like COCO for evaluating hallucinations, which may not be comprehensive or up-to-date. The use of more diverse evaluation benchmarks such as HallusionBench [3] could provide a more rigorous test of GACD's capabilities.
- The effectiveness of GACD may depend on the clarity and importance of visual information relative to the prompt, which could be a limitation in scenarios with complex or ambiguous visual inputs, such as documents and table images.

[1] https://internvl.github.io/blog/2024-07-02-InternVL-2.0/

[2] “Qwen2-VL: Enhancing Vision-Language Model's Perception of the World at Any Resolution.” ArXiv abs/2409.12191 (2024).

[3] “Hallusionbench: An Advanced Diagnostic Suite for Entangled Language Hallucination and Visual Illusion in Large Vision-Language Models.” 2024 IEEE/CVF Conference on Computer Vision and Pattern Recognition (CVPR)

**Questions:**

None

---

### Official Review · Reviewer_wQHe · 2024-11-01

**Soundness:** 3
**Presentation:** 3
**Contribution:** 4
**Rating:** 8
**Confidence:** 3

**Summary:**

The paper presents a novel decoding strategy, Gradient-based Influence-Aware Contrastive Decoding (GACD), designed to mitigate hallucinations in multi-modal large language models without the requirement for additional training. The authors identify three primary sources of hallucination: text-visual bias, co-occurrence bias, and long-term bias. To address these issues, they introduce an innovative technique that balances these biases by utilizing token influence through self-reflective gradient calculations. Notably, their approach tackles co-occurrence bias without necessitating further fine-tuning. Comprehensive experiments reveal that GACD not only effectively reduces hallucination but also achieves superior performance across various multi-modal benchmarks, outperforming existing decoding strategies.

**Strengths:**

1. The concept of employing gradient estimation to mitigate hallucination is innovative.

2. Empirical studies on token influence, encompassing text-visual bias, co-occurrence bias, and long-term bias, are solid and insightful.

3. Extensive experiments across various hallucination-related benchmarks underscore the effectiveness of the proposed approach on hallucination reduction.

**Weaknesses:**

The experimental results perform the performance of GACD on hallucination-related datasets which is not sufficient to show the generalization of proposed method. The authors need to show the model's performance with GACD on comprehensive benchmarks such as MMVet [1], MMBench [2], or MMMU [3].

[1] Yu, Weihao, et al. "Mm-vet: Evaluating large multimodal models for integrated capabilities." arXiv preprint arXiv:2308.02490 (2023).

[2] Liu, Yuan, et al. "Mmbench: Is your multi-modal model an all-around player?." European Conference on Computer Vision. Springer, Cham, 2025.

[3] Yue, Xiang, et al. "Mmmu: A massive multi-discipline multimodal understanding and reasoning benchmark for expert agi." Proceedings of the IEEE/CVF Conference on Computer Vision and Pattern Recognition. 2024.

**Questions:**

1. Is there any measurement of the computation like decoding time for GACD and other approaches such as Woodpecker[1], VCD[2], and AVISC[3]?

2. It is better to show the component analysis on other datasets such as POPE and AMBER to show the validness of each component.

[1] Yin, Shukang, et al. "Woodpecker: Hallucination correction for multimodal large language models." arXiv preprint arXiv:2310.16045 (2023).

[2] Leng, Sicong, et al. "Mitigating object hallucinations in large vision-language models through visual contrastive decoding." Proceedings of the IEEE/CVF Conference on Computer Vision and Pattern Recognition. 2024.

[3] Woo, Sangmin, et al. "Don't Miss the Forest for the Trees: Attentional Vision Calibration for Large Vision Language Models." arXiv preprint arXiv:2405.17820 (2024).

---

### Official Review · Reviewer_UUZj · 2024-11-03

**Soundness:** 2
**Presentation:** 2
**Contribution:** 3
**Rating:** 3
**Confidence:** 5

**Summary:**

This paper addresses the problem of hallucination in Multimodal Large Language Models (MLLMs), proposing a novel method called Gradient-based Influence-Aware Contrastive Decoding (GACD). The authors identify three biases that contribute to hallucinations: text-visual bias, co-occurrence bias, and long-term bias. The GACD method seeks to balance these biases by measuring "token influence" through gradient analysis. This approach allows the model to amplify relevant visual tokens while mitigating distractor tokens, particularly effective against co-occurrence hallucinations. Through extensive experiments, the paper demonstrates that GACD improves hallucination mitigation across multiple datasets and model types, outperforming state-of-the-art methods.

**Strengths:**

- The paper introduces the "token influence" method, which uses gradient norms to measure the sensitivity between input and output tokens. This analysis reveals the model's biases and sheds light on its internal behavior, especially regarding spurious correlations.
- The authors provide a CHAIR metric component analysis, which can be interpreted as evidence of mitigation for each bias—text-visual, co-occurrence, and long-term. This approach offers an indirect yet insightful perspective on the effectiveness of bias mitigation, supporting the method's unique contribution by showcasing its targeted impact on these specific biases.

**Weaknesses:**

- In Figure 1, the cause of hallucination appears ambiguous. The example of "food" doesn't strongly exhibit a text-visual bias, making it unclear why this specific example is used to demonstrate hallucination.
- The writing style is challenging to follow, particularly in the introduction and methodology sections. It has complex sentence structures, making it difficult to immediately grasp the author’s primary argument. Clearer, more straightforward language and improved structure would enhance readability.
- The paper does provide step-by-step explanations of individual components within the GACD method, such as token influence measurement and contrastive decoding adjustments. However, it lacks a clear, overarching view of how these steps fit together in the overall process. This makes it difficult to grasp how the entire method operates as a cohesive framework.
- In the experiment tables, particularly for the POPE dataset, the use of terms like "original" is ambiguous without clear definitions. Readers might not understand if this term refers to baseline results or some other metric. More specific labels or definitions would help clarify these results.
- For the LLava-QA90 and CHAIR experiments, the baselines and comparative methods are poorly defined. Each approach seems to have different parameters, which reduces the effectiveness of comparisons in showing efficiency or the superiority of the proposed method. Clearly stating consistent baselines and providing more context for each comparison would improve interpretability.

**Questions:**

- Could you provide more details on why the Manhattan (L1) norm was specifically chosen for token influence analysis? Have other norms been tested, and if so, how did they compare?
- The choice of early stopping thresholds (e.g., 7% for LLaVA, 25% for InstructBLIP) lacks detailed justification. Could you provide insights into how these values were determined and if they are consistent across different models?

---

### Note · Authors · 2024-11-14

I have read and agree with the venue's withdrawal policy on behalf of myself and my co-authors.